# Phospholipids, Sphingolipids, and Cholesterol-Derived Lipid Mediators and Their Role in Neurological Disorders

**DOI:** 10.3390/ijms251910672

**Published:** 2024-10-03

**Authors:** Akhlaq A. Farooqui, Tahira Farooqui

**Affiliations:** Department of Molecular and Cellular Biochemistry, The Ohio State University, Columbus, OH 43210, USA; akhlaqfarooqui@gmail.com

**Keywords:** phospholipids, arachidonic acid, docosahexaenoic acid, prostaglandins, leukotrienes, thromboxane, lipoxins, resolvins, protectins, maresins, isoprostane, neuroprostane, 4-hydroxynonal, 4-hydroxyhexanal, ceramide-1-phosphate, sphingosine, sphingosine 1-phosphate, hydroxycholesterol, 7-ketocholesterol

## Abstract

Neural membranes are composed of phospholipids, sphingolipids, cholesterol, and proteins. In response to cell stimulation or injury, the metabolism of lipids generates various lipid mediators, which perform many cellular functions. Thus, phospholipids release arachidonic acid or docosahexaenoic acid from the sn-2 position of the glycerol moiety by the action of phospholipases A_2_. Arachidonic acid is a precursor for prostaglandins, leukotrienes, thromboxane, and lipoxins. Among these mediators, prostaglandins, leukotrienes, and thromboxane produce neuroinflammation. In contrast, lipoxins produce anti-inflammatory and pro-resolving effects. Prostaglandins, leukotrienes, and thromboxane are also involved in cell proliferation, differentiation, blood clotting, and blood vessel permeability. In contrast, DHA-derived lipid mediators are called specialized pro-resolving lipid metabolites (SPMs). They include resolvins, protectins, and maresins. These mediators regulate immune function by producing anti-inflammatory, pro-resolving, and cell protective effects. Sphingolipid-derived metabolites are ceramide, ceramide1-phosphate, sphingosine, and sphingosine 1 phosphate. They regulate many cellular processes, including enzyme activities, cell migration and adhesion, inflammation, and immunity. Cholesterol is metabolized into hydroxycholesterols and 7-ketocholesterol, which not only disrupts membrane fluidity, but also promotes inflammation, oxidative stress, and apoptosis. These processes lead to cellular damage.

## 1. Introduction

“Lipid mediators” are chemical messengers which are produced locally through specific biosynthetic pathways in response to either neural cell activation or neuronal injury. Lipid mediators are lipophilic molecules which produce their effect either by binding to their receptors or by inducing inflammation and oxidative stress [1]. They not only play important roles in internal and external communication, but also modulate cellular growth, differentiation, adhesion, and migration [2]. In membranes, lipids are organized in bilayers, with the amine-containing phospholipids enriched on the cytoplasmic side of the plasma membrane (PM), while the choline-containing phospholipids and sphingolipids are enriched on the outer surface. The two lipid bilayers of membranes are held together by hydrophobic, coulombic, and van der Waal forces and hydrogen bonding [2]. This organization of the lipid bilayer is spontaneous, meaning it is a natural process which does not require energy. The distribution of phospholipids and sphingolipids in the two leaflets of the lipid bilayer is asymmetric. Phospholipids and sphingolipids contribute to the lipid asymmetry, whereas cholesterol and sphingolipids form lipid microdomains or lipid rafts that float within the membranes along with proteins. A large number of signaling molecules are concentrated within lipid rafts, which function as signaling centers capable of facilitating efficient and specific signal transduction pathways [3]. The interactions of an agonist with its receptors on the neural membrane surface results in the enhancement of phospholipid, sphingolipid, and cholesterol metabolism. This process not only increases the activities of phospholipid, sphingolipid, and cholesterol metabolizing enzymes and increases levels of lipid mediators, but also modulates many physicochemical properties of neural membranes such as fluidity, lateral pressure profile, bilayer thickness, permeability, and activity of ion channels [4]. Thus, the lipid metabolism in the brain is a tightly regulated process. Any alteration/dysregulation of the lipid metabolism may impact brain health and functions. In the brain, the function of the signal transduction network is to convey extracellular signals from the neural cell surface to the nucleus, where lipid mediators mediate biological responses at the gene level [5]. The dysregulation of the lipid mediator metabolism has been linked to neuroinflammation, oxidative stress, and apoptotic cell death in neurological disorders [6].

## 2. ARA-Derived Lipid Mediators

In phospholipids, arachidonic acid (ARA, 20:4n-6) and docosahexaenoic acid (DHA; 22:6n-3) are located at the sn-2 position of the glycerol moiety [7]. The majority of ARA is enriched in phosphatidylcholine (PtdCho), whereas phosphatidylethanolamine (PtdEtn) and ethanolamine plasmelogen (PlsEtn) contain both ARA and DHA. Phosphatidylserine (PtdSer) is enriched in DHA [7]. Cytosolic phospholipase A_2_ (cPLA_2_) liberates ARA from phospholipids, whereas DHA is released by the action of Ca^2+^-independent PLA_2_ [2] (Figure 1). Under physiological conditions, some ARA is oxidized by cyclooxygenases (COXs) and lipoxygenases (LOXs). These enzymes transform ARA into pro-inflammatory prostaglandins (PGs), leukotrienes (LTs), thromboxane (TX), and an anti-inflammatory lipoxins [2,8,9] (Figure 1). The other product of the PLA_2_-catalyzed reaction is lysophospholipid. This metabolite is the immediate precursor of platelet-activating factor, a potent inflammatory mediator. It produces it effects by binding to platelet-activating factor receptors [4,7]. The accumulation of lysophospholipids is controlled either through reacylation to native phospholipids [7] or by their metabolism into water-soluble glycerophosphodiesters such as glycerophosphocholine by lysophospholipases [2].

PGs are potent autocrine and paracrine lipid mediators which play an important role in physiologic and pathophysiologic responses in the brain. Among the 12 PGs, the most potent are PGD2, PGE2, and PGF2. PGE2 mediates its signaling through four distinct G-protein-coupled receptors, EP1, EP2, EP3, and EP4, which are encoded by different genes and differ in their responses to various agonists and antagonists and are differentially expressed on neuronal and glial cells throughout the brain (Figure 1). In addition, the brain also contains the PGF receptor (FP), the PGI receptor (IP), and the TxA receptor (TP) [10,11,12]. These lipid mediators play important roles in neurotransmitter release, sleep, and vasodilation and vasoconstriction of cerebral vessels [4,7].

Lipoxygenases (5-LOX, 12-LOX, and 15-LOX) are non-heme, iron-containing dioxygenases that insert molecular oxygen into ARA [9,13]. Five LTs, namely leukotriene A4 (LTA4), leukotriene B4 (LTB4), leukotriene C4 (LTC4), leukotriene D4 (LTD4), and leukotriene E4 (LTE4), are synthesized from ARA in various body tissue. LTA4 and LTB4 (non-cysteinyl leukotrienes) are structurally different from the cysteinyl leukotrienes (Cys-LT) as they lack the cysteine moiety, which is present in the Cys-LT (LTC4, LTD4, and LTE4) [13,14]. These leukotrienes can interact with BLT1 and BLT2 receptors [15], whereas LTC4, LTD4, and LTE4 are the ligands for the majority of cysteinyl leukotriene type 1 (CysLT1R) and type 2 receptors (CysLT2R) [16].

Thromboxanes (TXs) are synthesized from ARA by the sequential action of three enzymes—cPLA_2_, COX-2, and TXA2 synthase (TXAS) [17]. TXs are not only potent hypertensive agents, but also play an important role in platelet aggregation. Among thromboxanes, TXA2 interacts with thromboxane receptors (TPs) [18], which are linked with G-protein-linked receptors. Levels of thromboxane are elevated in cardiovascular, cerebrovascular, and inflammatory visceral diseases [19]. The irreversible inhibition of TXA2 with low-dose aspirin is currently used as an antiplatelet therapy for the prevention of primary and secondary vascular thrombotic events. TXA2 also plays an important role in vasoconstriction, adhesion molecule expression, inflammation, cell migration, proliferation, and hypertrophy [19,20].

LXs are classified into two groups: native LXs and aspirin-triggered lipoxins (ATLs). LXs include lipoxin A4 and lipoxin B4. ATLs include aspirin-triggered lipoxin A4 (15-epi-LXA4, ATLA4) and aspirin-triggered lipoxin B4 (15-epi-LXB4, ATLB4). Compared to native LXs, ATLs are more resistant to metabolic inactivation and have an enhanced ability to evoke bioactions. LXs are involved in the resolution of inflammation and stimulation of non-phlogistic phagocytosis of apoptotic cells by microglial cells [9]. In addition, LXs also produce antioxidative, antiapoptotic, and autophagy-moderating effects [21,22,23,24].

Other endogenous lipid mediators of the fatty acid metabolism are fatty acid ethanolamides (arachidonylethanolamide (AEA) and palmitoylethanolamide (PEA)). These lipid mediators play a primary role in inhibiting chronic inflammation in many neurodegenerative diseases [6,7,8]. The molecular mechanisms of the action of PEA in the brain are not fully understood. However, it is proposed that PEA exerts its neuroprotective effects through three mechanisms. The first mechanism suggests that PEA acts by down-regulating mast-cell degranulation, via an amide antagonist; the second mechanism postulates that PEA acts by enhancing anti-inflammatory and anti-nociceptive effects; and the third mechanism involves the direct stimulation of either PPAR-γ or the orphan receptor G-protein coupling GPR55 via the production of many anti-inflammatory effects by PEA [6,7,8].

## 3. Non-Enzymatic Oxidation of ARA

The non-enzymatic oxidation of ARA results in the generation of various metabolites such as 4-hydroxy 2-nonenal (4-HNE), reactive oxygen species (ROS), isoprostanes (IsoPs), isoketals (IsoKs), isofurans (IsoFs), malondialdehyde (MDA), and acrolein (Ac) [4,7] (Figure 1). Among these mediators, 4-HNE is a nine-carbon α, β-unsaturated aldehyde containing three functional groups. It is prone to be attacked by nucleophiles such as thiol or amino groups in proteins. It is highly toxic and an important biomarker for oxidative stress [25,26]. 4-HNE differentially modulates cell death, growth, and differentiation. The detoxification of 4-HNE involves conjugation with glutathione. Lower intracellular concentrations (<2 µM) of 4-HNE produce beneficial effects in cells by promoting cell survival and proliferation [26]. However, at higher concentrations, 4-HNE (10 to 60 μM) produces genotoxic effects by producing sister chromatid exchange, micronuclei formation, and DNA fragmentation. Furthermore, 4- HNE at (>100 μM) inhibits enzymes of glycolysis, mitochondrial respiration, DNA metabolism, and protein synthesis [26]. In addition, 4-HNE also inhibits many enzymes such as MAP kinase, caspases, ATPase, and enzymes of cell cycle [27,28]. 4-HNE also regulates transcription factors that are responsible for redox homeostasis (Ref-1, Nrf2, p53, NFκB, and Hsf1). Levels of 4-HNE are increased in stroke, Alzheimer’s disease (AD), Parkinson’s disease (PD), amyotrophic lateral sclerosis (ALS), and prion disease [29,30,31].

## 4. Other Non-Enzymatic Metabolites of ARA Metabolism

Non-enzymatic oxidation of ARA produces isoprostanes (IsoPs), isoketals (IsoKs), isofurans (IsoFs), malondialdehyde (MDA), acrolein (Ac), (Figure 2) and reactive oxygen species (ROS). Among the above lipid mediators, IsoPs are not only potent vasoconstrictors [32,33], but also contribute to cell proliferation, mitogenesis, and monocytic adhesion. These processes may be closely related to the onset of inflammation and atherosclerosis in the body [34,35,36].

The formation of IsoKs also occurs through the rearrangement of H2-IsoP endoperoxides (Figure 2). IsoKs are highly reactive γ-ketoaldehydes that form pyrrole adducts with the ε-amino group of lysine residues on protein in tissues and biological fluids [37]. IsoKs inhibit the activity of proteasomes in glial cells with an IC50 of 330 nM and induce cell death with an IC50 of 670 nM. Intra-hemispheric injections of 15-E2-IsoK disrupt the blood–brain barrier. 

Lipid peroxidation of unsaturated lipids produces lipid hydroperoxides (LOOH) [38]. Malondialdehyde (MDA) is the principal product of lipid peroxidation of polyunsaturated fatty acids (Figure 2). This aldehyde is highly toxic and is considered a biomarker for lipid peroxidation [39]. Its interaction with DNA and proteins has often been referred to as potentially mutagenic and atherogenic [39]. Acrolein is another metabolite of lipid peroxidation. It is the simplest α, β-unsaturated aldehyde which can form Michael adducts with thiol groups of cysteines that affect the activity of many proteins. Acrolein can readily enter the cell, where it causes glutathione depletion and thus can break the cellular redox balance and subsequently lead to oxidative stress [40]. Acrolein promotes apoptosis and adducts accumulate in several pathological conditions [41]. 

The non-enzymic oxidation of ARA also produces reactive oxygen species (ROS), which are the primary effector molecules of oxidative stress. ROS are produced under physiological states as well as in pathological conditions (AD, PD, ALS) and Huntington’s disease (HD). At low levels, ROS act as signaling molecules and regulate fundamental processes such as cell growth and adaptation responses [41,42]. However, high levels of ROS produce oxidative stress. This process impairs molecular signaling pathways, alters enzyme activities, and damages cellular lipids, proteins, or DNA leading to metabolic alterations in pathological conditions such as AD, PD, ALS, and HD [43,44].

## 5. DHA-Derived Lipid Mediators

Free DHA is enzymatically metabolized into resolvins (RVs), protectin D1 (PD1), and maresins (MaRs) (Figure 3). These mediators induce anti-inflammatory and pro-resolutionary effects [45]. In the presence of aspirin, the action of COX-2 on DHA produces aspirin-triggered forms of RVs that not only produce potent anti-inflammatory and immunoregulatory effects, but also regulate leukocyte trafficking [46,47]. RVs also reduce cytokine expression in isolated microglial cells [48].

DHA and its oxygenated derivatives are endogenous ligands for retinoid X receptors (RXRs) and PPAR receptors [49,50,51,52] in different cell types, including neurons and astrocytes [53,54,55]. Converging evidence suggests that DHA and its metabolites modulate the expression of a number of genes which control inflammation, cell survival, DNA binding, transcriptional regulation, transport, cell adhesion, cell proliferation, and raft formation [56,57].

## 6. Non-Enzymatic Oxidation of DHA

The non-enzymatic oxidation of DHA generates 4-hydroxyhexanal (4-HHE), neuroprostane (NP), neuroketal (NK), and neurofuran (NF) (Figure 2) [58]. DHA is highly enriched in neurons, so the generation of these metabolites has been used as an important index of neuronal protection/damage. The generation of 4-HHE from DHA results in stimulation of the Keap1-Nrf2 pathway. This process results in cardioprotective effects of DHA. In contrast, ARA produces 4-HNE, a lipid mediator, which produces oxidative stress [59]. NPs have 22 carbons and 4 double bonds and are analogous to IsoPs. The synthesis of NPs from DHA also produces peroxyl radicals, which may contribute to alterations in neural membrane fluidity and permeability leading to neuronal dysfunction [58].

## 7. Sphingolipid-Derived Lipid Mediators

Sphingomyelin (SM) is a major sphingolipid of myelin sheaths of neurons and lipid rafts in neural cell membranes. SM is hydrolyzed by sphingomyelinase (SMase), an enzyme that produces ceramide (Cer) and choline (Figure 4). Three SMases are known to occur in the brain, namely acid SMase (aSMase), neutral SMase, and alkaline SMase [60,61]. Ceramide consists of a sphingosine backbone attached to fatty acids (palmitic (C16) and stearic (C18) non-hydroxy fatty acids) by an amide bond. In the brain, ceramide synthesis not only occurs through de novo synthesis in the endoplasmic reticulum, but also via SMases [60,61,62]. From the endoplasmic reticulum, ceramide is transported by the ceramide transport protein (CERT) to the Golgi apparatus, where it is required for the synthesis of sphingomyelin (CerPCho or SM) [63,64] (Figure 4). Ceramide functions as a second messenger in a variety of cellular events, including proliferation, differentiation, growth arrest, inflammation, stress responses, synaptic activity, and apoptosis [65,66,67,68,69]. Ceramide is phosphorylated into ceramide1 phosphate (C 1-P) by ceramide kinase (Figure 4) [70]. C1-P activates cPLA_2_, an enzyme which hydrolyzes PtdCho and generates lyso-PtdCho and ARA, a fatty acid, which is converted into pro-inflammatory PGs, LTs, and TXs. Ceramide also activates serine/threonine protein kinases and phosphatases [71,72]. The crosstalk between sphingolipids metabolites and transcription factors (NF-κB and FOXOs) may be important for immune regulation and cell survival/death [73]. Stimulation of the atypical protein kinase zeta (PKCζ) by ceramide results in the suppression of mitogenesis [74,75]. In contrast, C 1-P stimulates cell migration, proliferation, angiogenesis, cell survival, and metabolism [76].

The degradation of ceramide by ceramidase results in the synthesis of sphingosine, which is then phosphorylated by ATP in the presence of sphingosine kinases (SphKs). This reaction results in the synthesis of sphingosine 1 phosphate (S1P) [77]. This metabolite promotes inflammation, cell proliferation, cell survival, and angiogenesis. It also contributes to neuritogenesis and immune function [78,79]. S1P can be converted back to sphingosine by S1P phosphatases (SPPase) or can be irreversibly broken down by sphingosine phosphate lyase (SPL) [78,79]. Collective evidence suggests that sphingolipid metabolites regulate diverse processes, including cell survival, oxidative stress, inflammation, apoptosis, and proliferation.

## 8. Cholesterol-Derived Lipid Mediators

Among various body organs, the brain is the richest source of cholesterol in the body. Two pools of cholesterol are present in the brain. One pool, which is metabolically stable, accounts for ∼70% of the total cholesterol. This pool is present in the myelin membranes of white matter [80]. The second pool, which represents ∼30% of the total cholesterol, is associated with the plasma and subcellular membranes of neurons and glial cells of gray matter. This is metabolically active and contributes to the formation of lipid rafts. Free cholesterol cannot cross the BBB. However, some cholesterol oxidation products (27-hydroxycholesterol, 24S-hydroxycholesterol, and 7-ketocholesterol) may diffuse into the brain [81].

Hydroxycholesterols are not only important regulators of cholesterol metabolism and lipid homeostasis but also play important role in immune function and membrane fluidity regulation. In the liver, the activation of some nuclear receptors such as liver X receptor α (LXRα) and RAR-related orphan receptors by hydroxycholesterols results in the regulation of various physiological processes in multiple tissues (Figure 5) [82]. These cholesterol metabolites not only produce strong pro-apoptotic and pro-inflammatory effects [83,84], but also impact on the renin–angiotensin system [85,86]. Among cholesterol metabolites, 27-Hydroxycholesterol depletes glutathione, promotes the generation of ROS, and induces inflammation and apoptosis [87]. 7-Ketocholesterol (7-KC) is formed during ROS attack on the carbon 7 of cholesterol (Figure 5). Increased levels of 7-KC have been found in the tissues, plasma, and/or cerebrospinal fluid of patients with major age-related diseases (cardiovascular diseases, eye diseases, and neurodegenerative diseases) [88]. 7-KC not only promotes an increase in Ca^2+^ but also activates cPLA_2_, an enzyme that releases ARA. This ARA interacts with 7-KC in the presence of Acyl-CoA cholesterol acyltransferase (ACAT) to form the 7KC-ARA complex [89]. It is suggested that 7-KC damages vascular endothelial cells by inducing inflammatory responses. This process elevates the risk of cardiovascular diseases, AD, and age-related macular degeneration. In addition, unesterified 7-KC not only disrupts membrane fluidity but also promotes inflammation, oxidative stress, and apoptosis. In the retina, 7- KC increases retinal microglial cell migration and angiogenicity. These processes may be involved in age-related macular degeneration [90,91,92]. 

## 9. Involvement of Lipid Mediators in Neurological Disorders

In the brain tissue, phospholipids, sphingolipids, and cholesterol are not merely structural components, but important molecules which play crucial roles in the maintenance of membrane fluidity, permeability, and membrane functionality modulating neurotransmitter release and receptor activity. These lipids also play significant roles in synaptic function, aiding in the formation and maintenance of synapses, as well as in synaptic plasticity, which is essential for learning and memory. Cholesterol, for example, is a key component of the myelin sheath, facilitating rapid signal transmission along neuronal axons [2,4]. Receptor-mediated degradation of neural membrane phospholipids, sphingolipids, and cholesterol by phospholipase A_2_, cyclooxygenase, lipoxygenase, acyltransferase, sphingomyelinase, and cytochrome P450 results in the generation of enzymic and non-enzymic lipid mediators of phospholipid metabolism. Enzymically derived lipid mediators promote neuroinflammation (5). In contrast, an increase in the non-enzymic lipid mediators of the phospholipid metabolism results in an elevated production of ROS, causing the onset of oxidative stress (6). Both of these processes are closely interrelated and are responsible for the pathogenesis of different neurodegenerative and neuroinflammatory disorders such as stroke, AD, PD, HD, and ALS [4,6]. In the brain, the metabolism of phospholipid-, sphingolipid-, and cholesterol-derived lipid mediators is an interrelated and interconnected process [5]. Thus, many cellular stimuli (neurotransmitters, cytokines, chemokines, and growth factors) modulate more than one enzyme of phospholipid, sphingolipid, and cholesterol metabolism at the same time [5]. 

The treatment of cells with exogenous sphingolipids results in the siphoning of cholesterol from the plasma membrane, suggesting that perturbations in sphingolipid levels are coupled with changes in cholesterol metabolism [93]. It has been reported that 25-hydroxycholesterol produces a significant increase in SM synthesis, which is dependent on oxysterol binding protein (OSBP), ceramide transport protein (CERT), and their shared binding partner VAP [93]. The precise mechanism of this process is not fully understood, but OSBP appears to activate CERT by promoting its recruitment to membranes and its binding to vesicle-associated membrane protein-associated protein (VAP). Under physiological conditions, the homeostasis among enzymes of phospholipid, sphingolipid, and cholesterol metabolism is based not only on optimal levels of lipid mediators, but also on the complexity and interconnectedness of signal transduction processes involved in their metabolism. However, in neurological disorders, such as stroke, Alzheimer’s disease (AD), Parkinson’s disease (PD), Huntington’s disease (HD), and amyotrophic lateral sclerosis (ALS), a marked increase in levels of lipid mediators may disturb cellular homeostasis. In addition, this process may also prevent the crosstalk among phospholipid-, sphingolipid-, and cholesterol-derived lipid mediators. This may result in a lack of communication among neurons, astroglia, and microglia. Furthermore, the increase in lipid mediator levels may also promote the progression of neurological disorders by controlling the oligomerization of aggregate pathogenic proteins (β-amyloid (Aβ), α-synuclein (α-Syn), mutated huntingtin (Htt), and mutated Cu/Zn-superoxide dismutase1 (SOD1)) associated with the pathogenesis of each disease. These processes may not only threaten the integrity of neural cell lipid homeostasis but also facilitate neurodegeneration in neurological disorders [2,6,7].

Recent studies on the pharmacological effects of PEA and luteolin (a polyphenol) in the brain have indicated that these molecules produce similar effects. Combined administration of PEA and luteolin inhibits the two main conspirators of neurodegenerative diseases, low-grade inflammation and oxidative stress, supporting the view that joint treatment using PEA plus luteolin has a superior effect compared to the molecules used alone by stimulating both hippocampal neurogenesis and dendritic spine maturation [94,95]. In particular, the association of these two molecules has been evaluated in a lot of different experimental models, such as in AD and PD, vascular dementia, anxiety and depression, brain and spinal cord injury, and arthritis [94,95]. In addition, these studies provide strong evidence that the combination of PEA with luteolin is able to significantly decrease the neuroinflammatory and apoptotic pathways, modulating the release of cytokines and the activation of astrocytes and microglia, decreasing oxidative and nitrosative stress, and is able to enhance the expression of the neurotrophic factors promoting neuronal regeneration as well as demonstrate that this association possesses the ability to modulate the autophagic process [94,95].

Collective evidence suggests that the dysregulation of the balance among phospholipids, sphingolipids, and cholesterol may results in “lipo-toxicity” associated with the pathophysiology of common metabolic and neurodegenerative diseases, typically characterized not only by increased ceramide/sphingosine pools but also by the induction of inflammation and oxidative stress. Lipo-toxicity is closely associated with cell death involving inflammation and oxidative stress [96,97]. The severity of the lipo-toxic insult can be modulated by the specific cellular genetic vulnerability to the toxicity induced by phospholipid and sphingolipid metabolites.

## 10. The Inborn Errors of Sphingolipid Metabolism

The accumulation of sphingolipids and their lipid mediators also occurs in lysosomal storage diseases [98,99]. These diseases are collectively known as sphingolipidoses. In these pathological conditions, the accumulation of sphingolipids may not only impact neuronal and myelin membrane stability and lipid homeostasis, but also promote the onset of neurodegeneration [99]. Studies on human autopsy brains and brain tissue from animal models of sphingolipidoses have indicated that the accumulation of sphingolipids may be due to the deficiency of lysosomal enzymes involved in the degradation of sphingolipids [98,99]. Thus, Niemann–Pick disease (NP), Gaucher’s disease (GD), Farber disease (FARD), Fabry disease (FD), and Krabbe’s disease (KD) are caused by the deficiency of sphingomyelinase, beta-glucocerebrosidase, ceramidase, alpha-galactosidase, and beta-galactosidase, respectively, with the accumulation of sphingomyelin, glucosylceramide, ceramide, trihexosylceramide, and galactosylceramide (Table 1).

In NP, the deficiency of aSMase produces a defect in SM degradation which means that SM cannot be converted to Cer, and consequently, the Cer to SM ratio is altered. It means that signal transduction processes involving ceramide are not developed, mediating changes in brain metabolism [100,101]. In GD, the accumulation of glucosylceramide and glucosylsphingosine produces a defect in the neuronal storage of gangliosides, leading to loss of neurons and their axons. This process may result in cortical atrophy and white matter degeneration [102]. In FARD, the deficiency of acid ceramidase results in high levels of ceramide [103]. The rate of Cer synthesis in brains of FARD patients is normal but the accumulation of ceramide and complex sphingolipids in the lysosomal compartment may cause abnormal Cer levels in the brain, resulting in neuronal dysfunction [104,105]. Similarly, in KD, the deficiency of β-galactosidase in the lysosomes results in the accumulation of galactocerebroside (GalCer) and galactosylsphingosine (GalSph) or psychosine. The accumulation of cytotoxic psychosine produces apoptotic cell death in oligodendrocytes, resulting in the demyelination of the brain and spinal cord [106,107]. It must be kept in mind that death in patients with sphingolipidoses occurs early in life (young age), but in AD, PD, and ALS, patients die in old age, even in the familial form of these diseases. This may be due to the involvement of different genes in sphingolipidoses and AD, PD, and ALS. Collective evidence suggests that defective sphingolipid metabolism is not only involved in sphingolipidoses, but also in AD, where sphingolipids are involved in the abnormal processing and aggregation of Aβ (the major components of senile plaques), which promote the induction of TNF-α, a pro-inflammatory cytokine. This cytokine is considered the main inducer of AD [108]. The neuronal plasma membrane is a primary target for Aβ and its lipid component is directly involved in Aβ neurotoxicity mechanisms. The most important conformational changes of Aβ occur in the presence of the sphingolipid V3-like domain of Aβ. This domain interacts with SM and galactosylceramide in monomolecular films at the air–water interface [109]. These changes in the membrane composition may also influence the activity of enzymes involved in APP processing.

Several studies have indicated that ceramide and monohexosylceramide metabolism are involved in the pathophysiology of PD [110]. It is also reported that gene mutations in glucocerebrosidase may be closely associated with the pathogenesis of PD [110]. Sphingolipids have been shown to promote aggregation of the monomeric form of α-synuclein (a protein associated with PD pathogenesis) into an oligomeric form (aggregated form), which may be a toxic form. 

**Table 1 ijms-25-10672-t001:** Sphingolipidosis caused by the deficiency of lysosomal enzymes.

Sphingolipidoses (Type)	Accumulated Sphingolipid	Deficient Enzyme	References
Farbar disease (FARD, acid ceramide deficiency)	Ceramide	Ceramidase	[101,103]
Gaucher disease	Glucosylceramide	Beta-gluco-cerebrosidase	[99,100]
Niemann-Pick disease	Sphingomyelin	Sphingomyelinase	[98]
Krabbe disease	Galactosylceramide	Beta-galactosidase	[111]
Fabry disease (Trihexosylceramide lipidosis)	Trihexosylceramide	Alpha-galactosidase	[112]

## 11. Sphingolipid Metabolites in Biofluids as Biomarkers for AD, PD, ALS, and Sphingolipidoses

As stated above, the pathophysiology of AD, PD, HD, and ALS involves the accumulation of Aβ, α-Syn, Huntingtin, and SOD1 in a selective subpopulation of neurons which undergo morphological and clinical changes [113]. An important feature of neurodegenerative diseases is an overproduction of the precursor protein associated with the formation of an aberrant protein with an unstable structure. This altered protein tends to form and accumulate intra- and extracellular protein clumps that may be related to neuronal death in the affected part of the brain [113]. Determination of the enzymatic activities associated with a deficiency of lysosomal enzymes and their substrates in the cerebrospinal fluid and blood of AD, PD, HD, and ALS may be helpful in identifying patients with neurodegenerative diseases. The main problems in developing ideal biomarkers have been the slow understanding of pathogenesis of AD, PD, HD, and ALS, and a lack of information on the progression and treatment of neurodegenerative diseases [114]. To determine and evaluate whether a treatment is working, it is of utmost importance to choose biomarkers that are most relevant and specific to neurodegenerative diseases [113]. At present, fingolimod (FTY720), an S-1-P receptor modulator, is the only drug undergoing clinical trials of phase II safety for the treatment of ALS [115]. Thus, the use of sphingolipid-derived metabolites as new diagnostic biomarkers and as targets for innovative therapeutic strategies in different neurodegenerative disorders is an important topic which should be thoroughly researched in animal models and human brain tissues.

## 12. Conclusions

Lipid mediators are important endogenous metabolites derived from enzymatic degradation of membrane lipids by PLA_2_, COXs, LOXs, acyltransferases, and SMases. Enzymatic oxidation of ARA produces pro-inflammatory PGs, LTs, and TXs. The non-enzymatic metabolites of ARA include 4-HNE, PEA, IsoP, IsoK, IsoF, MDA, Ac, and ROS. The latter interacts with NF-κB and promotes inflammation. In contrast, enzymatic oxidation of DHA generates anti-inflammatory lipid mediators (RVs, PDs, and MaR). These mediators directly or indirectly suppress the activity of NF-κB and inflammation. The sphingolipid-derived metabolites are ceramide, C 1-P, sphingosine, and S1P. These mediators are essential for cellular signaling. Their synthesis contributes to neuroinflammation, cell migration, and apoptosis. Cholesterol-derived mediators (hydroxycholesterols and 7-KC) are involved in neural cell differentiation, exocytosis, neuroinflammation, oxidative stress, and apoptosis. Levels of lipid mediators are markedly increased in neurological disorders. It still remains to be seen if there is an overlap in the neurochemistry of inborn errors of metabolism and AD, PD, and ALS because the onset of these pathological conditions occurs at very different ages.

## Figures and Tables

**Figure 1 ijms-25-10672-f001:**
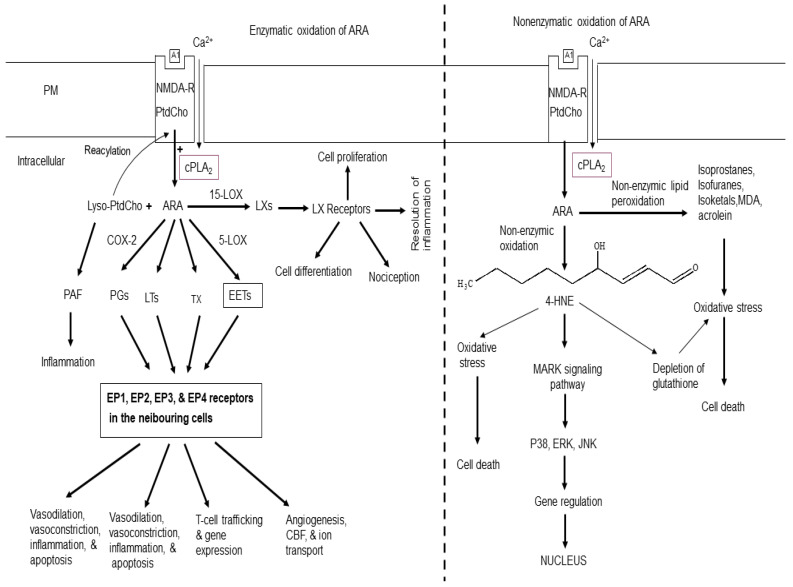
Enzymatic and non-enzymatic metabolism of arachidonic acid. Plasma membrane (PM); *N*-methyl-D-aspartate receptor (NMDA-R); glutamate (Glu); phosphatidylcholine (PtdCho); lysophosphatidylcholine (Lyso-PtdCho); cytosolic phospholipase A_2_ (cPLA_2_); arachidonic acid (ARA); cyclooxygenase-2 (COX-2); 5-lipoxygenase (5-LOX); 15-lipoxygenase (15-LOX); platelet-activating factor (PAF); epoxyeicosatetraenoic acids (EETs); prostaglandins (PGs); leukotrienes (LTs); thromboxane (TX); lipoxins (LXs); *4*-hydroxy 2-nonenal (4-HNE); cerebral blood flow (CBF); A1 (Glu); mitogen-activated protein kinase (P38); serine/threonine protein kinase (ERK).

**Figure 2 ijms-25-10672-f002:**
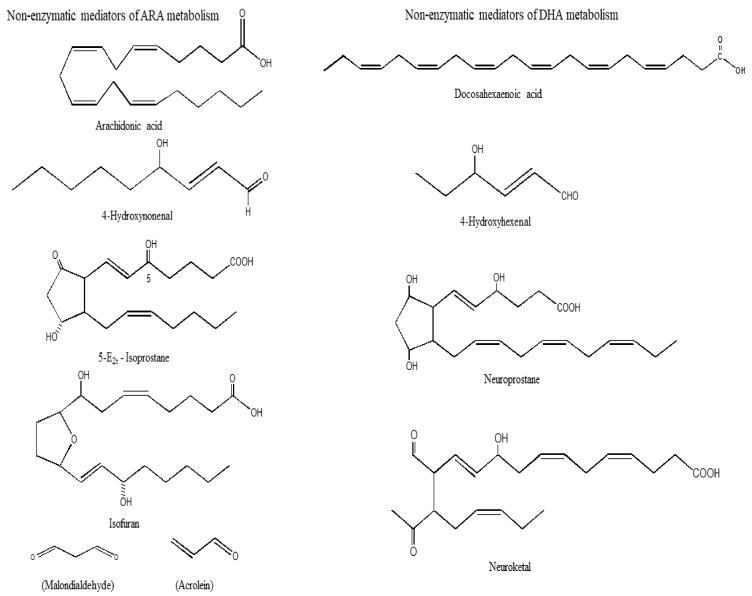
Chemical structures of non-enzymatic mediators of arachidonic acid metabolism.

**Figure 3 ijms-25-10672-f003:**
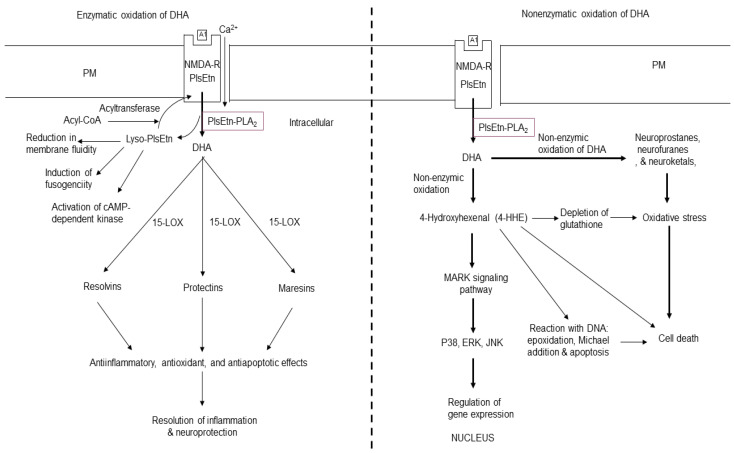
Enzymatic and non-enzymatic metabolism of docosahexaenoic acid. Plasma membrane (PM); N-methyl-D-aspartate receptor (NMDA-R); glutamate (Glu); ethanolamine plasmalogen (PlsEtn); lyso-plasmalogen (Lyso-PlsEtn); plasmalogen-specific phospholipase A2 (PlsEtn-PLA2); docosahexaenoic acid (DHA); 15-lipoxygenase (15-LOX); 4-hydroxyhexenal (4-HHE); A1 (Glu); mitogen-activated protein kinase (P38); serine/threonine protein kinase (ERK).

**Figure 4 ijms-25-10672-f004:**
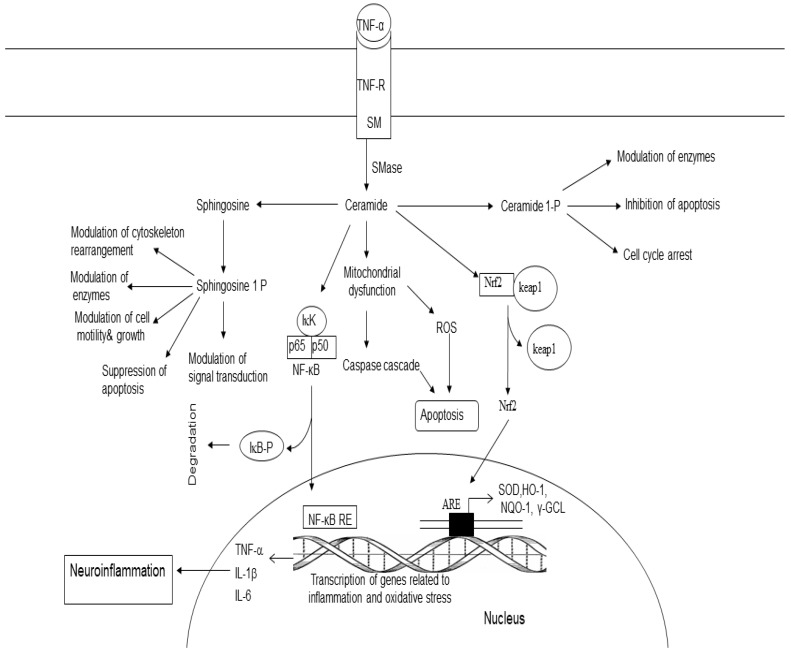
Metabolism and role of ceramide, ceramide 1-phosphate, and sphingosine 1 phosphate (S 1 P). Tumor necrosis factor-α (TNF-α tumor necrosis factor-α-receptor (TNF-R sphingomyelin (SM); sphingomyelinase (SMase); ceramide 1 phosphate (C1-P); nuclear factor-κB (NF-κB); nuclear factor-κB response element (NF-κB-RE); reactive oxygen species (ROS); tumor necrosis factor-α (TNF-α); interleukin-1β (IL-1β); interleukin-6 (IL-6); haemoxygenase (HO-1); NADPH quinine oxidoreductase (NQO-1); γ-glutamate cystein ligase (γ-GCL), and antioxidant response-element (ARE).

**Figure 5 ijms-25-10672-f005:**
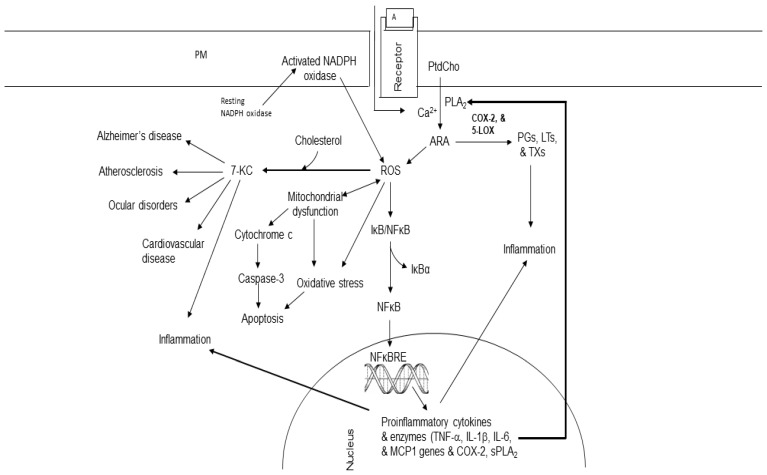
Effect of ROS on cholesterol and synthesis of 7-ketocholesterol in brain and visceral tissues. Plasma membrane (PM); *N*-methyl-D-aspartate receptor (NMDA-R); glutamate (Glu); phosphatidylcholine (PtdCho); lysophosphatidylcholine (Lyso-PtdCho); cytosolic phospholipase A_2_ (cPLA_2_); arachidonic acid (ARA); cyclooxygenase-2 (COX-2); 5-lipoxygenase (5-LOX); prostaglandins (PGs); leukotrienes (LTs); thromboxane (TX); nuclear factor-κB (NF-κB); nuclear factor-κB response element (NF-κB-RE); reactive oxygen species (ROS); 7-ketocholesterol (7-KC); tumor necrosis factor-alpha (TNF-α); interleukin 1β (IL-1β); interleukin-6 (IL-6); monocyte chemotactic protein-1 (MCP1); secretory phospholipase A_2_ (sPLA_2_).

## Data Availability

No new data were created or analyzed in this study.

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
