# Peer review of "Phospholipids, Sphingolipids, and Cholesterol-Derived Lipid Mediators and Their Role in Neurological Disorders"

_ijms, 2024, doi:10.3390/ijms251910672_

Round 1

Reviewer 1 Report (Previous Reviewer 1)

Comments and Suggestions for Authors

The authors addressed all my previously raised concerand and comments.

After carefully reading through the revised text I add couple of additional comments (see below).

Sentences from lines 371-388 do not belong to the sub-chapter 10, as thay are related to PD, not to inborn errors of sphingolypid metabolism, unless the authors are trying to establish a link between etiology and pathogenesis of Gaucher disease (disease of inborn errors of sphyngolypid metabolism) and PD. It this is the case, please add couple of sentences to make such link clear to reades.

I also suggest changing the title of sub-chapter 10 to "The inborn errors of sphingolypid metabolism" 

Author Response

Reviewer I:

Comment 1:

The authors addressed all my previously raised concerand and comments.

After carefully reading through the revised text I add couple of additional comments (see below).

Sentences from lines 371-388 do not belong to the sub-chapter 10, as thay are related to PD, not to inborn errors of sphingolypid metabolism, unless the authors are trying to establish a link between etiology and pathogenesis of Gaucher disease (disease of inborn errors of sphyngolypid metabolism) and PD. It this is the case, please add couple of sentences to make such link clear to reades.

Response of comment 1:

As suggested by reviewer 1, we have deleted text between line 371-388.

Comment 2:

I also suggest changing the title of sub-chapter 10 to "The inborn errors of sphingolypid metabolism" 

Response of comment 2: We have also changed the title of chapter of 10 to “The inborn errors of sphingolipid metabolism.

Reviewer 2 Report (New Reviewer)

Comments and Suggestions for Authors

I read with interest this paper which is bringing up important knowledge on the role fo new mediatos of phopsholipid, sphingolipids and cholesterol products in the field of brain diseases.

My only observatyion is the total lack of any refernece to the ole of this molecules in the field of neurodegenrative disease.

As authors surely know, in the recent years the field of neurodegenrative disease get enriched with new mediators of fatty acids, as for example the Palmitoylethanolamide which has been used in combination with Luteoline ion many neurodegenrative disorders (see as reference DOI 10.3390/biom12081161). this porduct is one of the most used also in clinical practice and it has gained major scientific relevance as mediator of th einflammatory processes contributing to the neurodegenration.

i would suggest authors to integrate with a little chapter from this molecule

Author Response

Reviewer II:

Comment 1:

I read with interest this paper which is bringing up important knowledge on the role fo new mediatos of phopsholipid, sphingolipids and cholesterol products in the field of brain diseases.

My only observatyion is the total lack of any refernece to the ole of this molecules in the field of neurodegenrative disease.

As authors surely know, in the recent years the field of neurodegenrative disease get enriched with new mediators of fatty acids, as for example the Palmitoylethanolamide which has been used in combination with Luteoline ion many neurodegenrative disorders (see as reference DOI 10.3390/biom12081161). this porduct is one of the most used also in clinical practice and it has gained major scientific relevance as mediator of th einflammatory processes contributing to the neurodegenration.

i would suggest authors to integrate with a little chapter from this molecule

Response to Reviewer II

As suggested by reviewer 2 we have added new information on N-arachidonoylethanolamine and its congener N-palmitoylethanolamine or palmitoylethanolamide (lines 118-128).

Other endogenous lipid mediators of phospholipid metabolism are fatty acid ethanolamides (arachidonylethanolamide and palmitoylethanolamide (PEA). These lipid mediator play a primary role in inhibiting the chronic inflammation in many neurodegenerative diseases [6, 7, 8]. The molecular mechanisms of the action of PEA in the brain is not fully understood. However, it is proposed that PEA exerts its neuroprotective effects through three mechanisms. The first mechanism advances that PEA acts by down-regulating mast-cell degranulation, via an amide antagonist; the second mechanism postulates that PEA acts by enhancing the anti-inflammatory and anti-nociceptive effects; and the third mechanism involves the direct stimulation of either PPAR-γ or the orphan receptor G-protein coupling, GPR55 by PEA’s producing many anti-inflammatory effects [6, 7, 8].

We have also included recently published information on the effects of PEA and luteolin in neurodegenerative diseases (lines 331-345).

Recent studies on pharmacological effects of PEA and luteolin (a polyphenol) in brain have indicated that these molecules produce similar effects. Combined administration of PEA and luteolin inhibits the two main conspirators of neurodegenerative diseases: low-grade inflammation and oxidative stress supporting the view that joint treatment using PEA plus luteolin has a superior effect compared to the molecules used alone by stimulating both hippocampal neurogenesis and dendritic spine maturation [94, 95].  In particular, the association of these two molecules has been evaluated in a lot of different experimental models, such as in AD and PD, vascular dementia, anxiety and depression, brain and spinal cord injury, and arthritis [94, 95]. In addition, these studies provide strong evidence that the combination of PEA with luteolin is able to significantly decrease the neuroinflammatory and apoptotic pathways, modulating the cytokines release and the activation of astrocytes and microglia, decreasing the oxidative and nitrosative stress, and is able to enhance the expression of the neurotrophic factors promoting neuronal regeneration, as well as demonstrate that this association possesses the ability to modulate the autophagic process [94, 95].

With above changes in the manuscript, we hope that you will find our manuscript acceptable for publication in IJMS.

References

  1. Landolfo E, Cutuli D, Petrosini L, Cattagirone C. Effects of palmitoylethanolamide on neurodegenerative

diseases: A review from rodents to humans. 2022; 12:667-684.

  1. Cordaro M, Cuzzocrea S, Crupi R. An update of palmitoylethanolamide and luteolin

Effects in preclinical and clinical studies of

neuroinflammatory events. Antioxidants. 2020;9: 216-244.

This manuscript is a resubmission of an earlier submission. The following is a list of the peer review reports and author responses from that submission.

Round 1

Reviewer 1 Report

Comments and Suggestions for Authors

Please find below my comments and suggestions.

Line 241 “Among various body tissues, brain is the richest source of cholesterol in the body” – brain is not a tissue, it’s an organ

Line 244. “The second pool, which represent” – represents

Line 249. “Hydroxycholesterol” – Hydroxycholesterols

Line 279. “sphingolipid-“ please fix the typo

Line 280. “which play crucial” - which play crucial role

Line 286. “sphingolipid-“ fix the typo

Lin 314. “ the increase in lipid mediator levels may also promote the progression of neurological disorders by controlling oligomerization of aggregate pathogenic proteins (Aβ, α-Syn, mutated huntingtin, and mutated Cu/Zn-superoxide dismutase1) associated with the pathogenesis of each disease”. What about pathological aggregation of lipids/sphingolipids, as a feature of such diseases?

What are the diagnostic/prognostic value and value as therapeutic targets of all the aforementioned mediators? Please provide more examples.   

The review is entitled “Phospholipids, sphingolipids, and cholesterol-derived lipid mediators and their role in neurological disorders”, however there is only one page describing role of the aforementioned mediators in neurological disorders.

I suggest significantly expanding discussion on this subject. There is a sub-set of neurological disorders with “inborn errors” of sphingolipid metabolism (hereditary diseases caused by pathological nucleotide variants in the genes encoding enzymes involved in sphingolipid metabolism), with symptoms of neurological disorders. Why do not authors characterize them in more detail? The authors may want to add several sub-chapters, focusing on Parkinson disease, Alzheimer Disease, Gaucher disease, Niemann–Pick disease, etc.

Reviewer 2 Report

Comments and Suggestions for Authors

The purpose of this review seems to simply remind the readers that there are several lipid mediators in the cells, and they play significant roles in health and disease. However, the manuscript is too simple, and its scope is limited. Discussion about Several prominent and key lipid mediators, involved in neurological disorders, are lacking.  The role of different enzymes including ceramide synthases, sPLA2, iPLA needs to be included. Similarly, receptors of PGs and leukotrienes, S1P etc. need to be described. Some other relevant comments are described below.

1.      The review focusses mainly on ARA and DHA-derived lipid mediators.  Other PUFA including 18:2, 18:3, 20:5 etc. are missing. In figure 1, other prominent reactive lipid aldehydes such as acrolein and MDA need to be included.

2.      Role of prominent brain lipids involved in many neurological diseases such as cerebrosides, sulfatides, sphingosine and psychosine is not included.

3.      Role of different fatty acid species of sphingomyelin and ceramide has not been described.

4.      Role of free fatty acids and their binding proteins needs to be included.

5.      Role of mitochondria-specific lipid mediator “cardiolipin” is not included. It plays very significant role to regulate ROS.

6.      There are several lipid-based vitamins such as Vitamins D, E A etc., are also not included.

Obviously, this review is incomplete and lacks several prominent aspects of lipid biology. As written and presented, the review does not add much to the knowledge of research scientists working in the field of lipid biology.